# SplineNets:
# Continuous Neural Decision Graphs

**Cem Keskin**
cemkeskin@google.com

Shahram Izadi
shahrami@google.com

## Abstract

We present SplineNets, a practical and novel approach for using conditioning in convolutional neural networks (CNNs). SplineNets are continuous generalizations of neural decision graphs, and they can dramatically reduce runtime complexity and computation costs of CNNs, while maintaining or even increasing accuracy. Functions of SplineNets are both dynamic (*i.e.*, conditioned on the input) and hierarchical (*i.e.*, conditioned on the computational path). SplineNets employ a unified loss function with a desired level of smoothness over both the network and decision parameters, while allowing for sparse activation of a subset of nodes for individual samples. In particular, we embed infinitely many function weights (*e.g.* filters) on smooth, low dimensional manifolds parameterized by compact B-splines, which are indexed by a position parameter. Instead of sampling from a categorical distribution to pick a branch, samples choose a continuous position to pick a function weight. We further show that by maximizing the mutual information between spline positions and class labels, the network can be optimally utilized and specialized for classification tasks. Experiments show that our approach can significantly increase the accuracy of ResNets with negligible cost in speed, matching the precision of a 110 level ResNet with a 32 level SplineNet.

## 1 Introduction and Related Work

There is a growing body of literature applying conditioning to CNNs, where only parts of the network are selectively active to save runtime complexity and compute. Approaches use conditioning to scale model complexity without an explosion of computational cost *e.g.* [1] or increase runtime efficiency by reducing model size and compute without degrading accuracy *e.g.* [2]. In this paper we present a new and practical approach for supporting conditional neural networks called SplineNets. We demonstrate how these novel networks dramatically reduce runtime complexity and computation cost beyond regular CNNs while maintaining or even increasing accuracy.

Conditional neural networks can be categorized into three main categories: (1) proxy objective methods that train decision parameters via a surrogate loss function, (2) probabilistic (mixture of experts) methods that assign scores to each branch and treat the loss as a weighted sum over all branches, and (3) feature augmentation methods that augment the representations with the scores.

The first group relies on non-differentiable decision functions and trains decision parameters using a proxy loss. Xiong *et al.* use a loss that maximizes distances between subclusters in their conditional convolutional network [3]. Baek *et al.* use purity of data activation according to the class label [4], and Biçici *et al.* use information gain as the proxy objective [5]. While the former approach only allows soft decisions, the latter method uses argmax of a scoring function to sparsely activate branches, which results in discontinuities in the loss function. Bulo *et al.* use multi-layered perceptrons as decisions, and the network otherwise acts like a decision tree [6]. Denoyer *et al.* train a tree structured network using the REINFORCE algorithm [7].

Ioannou *et al*. take a probabilistic approach, assigning weights to each branch and treating the loss as a weighted sum over losses of each branch [2]. They sparsify the decisions at test time, leading to gains in speed at the cost of accuracy. Shazeer *et al*. follow a similar probabilistic approach that assigns weights to a very large number of branches in a single layer [1]. They use the top-$k$ branches both at training and test time, leading to a discontinuous loss function. Another approach treats decision probabilities of a tree as the output of a separate neural network and then trains both models jointly, while slowly binarizing decisions [8]. However, inference still requires a regular network to be densely evaluated to generate the decision probabilities.

Finally, Wang *et al*. [9] take the approach of creating features from scores, enabling decisions to be trained via regular back-propagation. However, they have only decisions and do not build higher level representations.

Our method is novel and does not fit any of these categories. It is inspired from the recently proposed non-linear dimensionality reduction technique called Additive Component Analysis (ACA) [10]. This technique fits a smooth, low dimensional manifold to data and learns a mapping from it to the input space. We extend this work by learning both the projections onto these manifolds, as well as the mapping to the input space in a neural network setting. We then apply this technique to the function weights in our network. Hence, SplineNets employ subnetworks that explicitly form the main network parameters from some latent parameters, as in the case of Hypernetworks [11]. Unlike that work, we also project to these latent manifolds, conditioned on the feature maps. This makes SplineNet operations dynamic, which is similar to the approaches in Dynamic Filter Networks [12], Phase-functioned Neural Networks [13] and Spatial Transformer Networks [14]. We further condition these projections on previous projections in the network to make the model hierarchical. As these projections replace the common decision making process of selecting a child from a discrete set, this becomes a new paradigm for decision networks, which operates in continuous decision spaces. There is some similarity with Reinforcement Learning (RL) on Continuous Action Spaces [15], but the aim and problem setting for RL is much different from hierarchical decision graphs. For instance, rewards come from the environment in an RL setting, whereas decisions specialize the network and define the loss in our case.

The heart of SplineNets is the embedding of function weights on low dimensional manifolds described by B-splines. This effectively makes the branching factor of the underlying decision graph uncountable, and indexes the infinitely many choices with a continuous position parameter. Instead of making discrete choices, the network then chooses a position, which makes the decision process differentiable and inherently sparse. We call this the *Embedding Trick*, somewhat analogous to the *Reparameterization Trick* that makes the loss function differentiable with respect to the distribution parameters [16].

Load balancing is a common problem in hierarchical networks, which translates to under-utilization of splines in our continuous paradigm. The common solution of maximizing information gain based on label distributions on discrete nodes, as used in [5], translates to specializing spline sections to class labels or clusters in our case. We show that maximizing the mutual information between spline positions and labels solves both problems, and in the absence of labels, one can instead maximize the mutual information between spline positions and the input, in the style of InfoGANs [17].

Another common issue with conditional neural networks is that the mini-batch size gets smaller with each decision. This leads to serious difficulties with training due to noisier gradients towards the leaves and learning rate needing adaptation to each node, while also limiting the maximum possible depth of the network. Baek *et al*. use a decision jungle based architecture to tackle this problem [4]. Decision jungles allow nodes in a layer to co-parent children, essentially turning the tree into a directed acyclic graph [18]. SplineNets also do not suffer from this issue, since their architecture is essentially a decision jungle with an (uncountably) infinite branching factor. We further add a constraint on the range of valid children to simulate interesting architectures.

Our contributions in this paper are: (i) the embedding trick to enable smooth conditional branching, (ii) a novel neural decision graph that is uncountable and operates in continuous decision space, (iii) an architectural constraint to enforce a semantic relationship between latent parameters of functions in consecutive layers, (iv) a regularizer that maximizes mutual information to utilize and specialize splines, and (v) a new differentiable quantization method for estimating soft entropies.

## 2  Methodology

CNNs take an input $x^1{=}x$ and apply a series of parametric transformations, such that $x^{i+1}{=}T^i(x^i;\omega^i)$, where superscript $i$ is the level index. On the other hand, decision graphs typically navigate the input $x$ with decision functions parameterized by $\theta^i$, selecting an eligible node and the next parameters $\theta^{i+1}$ contained within. Skipping non-parametric functions for CNNs and assuming sparse activations for decision graphs, we can describe their behavior with the graphical models shown in Figure 1. Assuming a symmetrical architecture, such that $T^i_j$ for each node $j$ in layer $i$ has the same form but different weights, a *neural decision graph* (NDG) can then be described by the graphical model shown on the right. The decision process governed by parameters $\theta^i$ is used here to determine the next $\omega^i$ (red arrows) as well as $\theta^{i+1}$ (green arrows). Discrete NDGs then have a finite set of weights $\{\omega^i_j\}^{N^i}_{j=1} \in \mathbb{R}^{g^i}$ (and the corresponding $\{\theta^i_j\}^{N^i}_{j=1} \in \mathbb{R}^{h^i}$) to choose from at each level with $N^i$ nodes, depending on the features $x^i$ and choices from the previous level. In this work, we investigate the case where the set of choices is uncountably infinite and indexed by a real-valued position parameter, forming a continuous NDG.

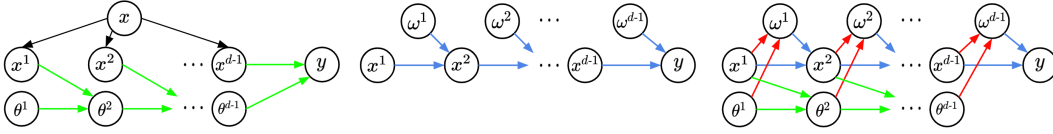

Figure 1: Graphical models: (left) Decision graph, (middle) CNN, (right) Neural decision graph.

### 2.1  Embedding trick

The embedding trick can be viewed as making the branching factor of the underlying NDG uncountably infinite, while imposing a topology on neighboring nodes. Each $\omega^i_j$ in the set of discrete choices is essentially a point in $\mathbb{R}^{g^i}$. In the limit of $N^i{\to}\infty$, these infinitely many points lie on a low dimensional manifold, as shown in Figure 2 (assuming $g^i{=}3$). In this work, we assume this manifold is a 1D curve in $\mathbb{R}^{g^i}$, parameterized by a position parameter $\phi^i \in [0,1]$, which is used to index the individual $\omega^i$'s. A simple example would be the set of all $5 \times 5$ edge filters, which form a closed 1D curve in a 25D space. Instead of making a discrete choice from a finite set, the network can then choose a $\phi^i$ using a smooth, differentiable function. The same process can be applied to the decision parameters $\theta^i_j$ to form another 1D curve in $\mathbb{R}^{h^i}$, indexed by the same position parameter $\phi^i$, since selecting a node $j$ should implicitly pick both $\omega^i_j$ and $\theta^i_j$ in an NDG.

We parameterize these latent curves with B-splines, since they have bounded support over their control points, or *knots*. This is a desirable property that allows efficient inference by allowing only a small set of knots be activated per sample, and it also makes training more well-behaved, since updates to control points can change the shape of the manifold only locally. Notably, this also efficiently solves the exploration problem, since the derivative of the loss with respect to the spline position exists, and the network knows which direction on the spline should further reduce the energy. The knots are trained offline, while the position parameters are dynamically determined during runtime.

While we only focus on 1D manifolds in this paper, our formulation allows easy extension to higher dimensional hypersurfaces in the same style as ACA, simply by describing surfaces as a sum of splines. This restricts the family of representable surfaces, but leads to a linear increase in the number of control points that need to be learned, as opposed to the general case where one needs exponentially many control points.

B–splines are piecewise polynomial parametric curves with bounded support and a desired level of smoothness up to $C^{d-1}$, where $d$ is the degree of the polynomials. The curve approximately interpolates a number of knots, such that $S(\phi) = \sum_{k=1}^{K} C_k B_k(\phi)$. Here, $C_k, k = 1..K$ are the knots and $B_k(\phi)$ are the basis functions, which are piecewise polynomials of the form $\sum_{t=0}^{d} a_t \phi^t$. Coefficients $a_t$ are fixed and can be determined from continuity and smoothness constraints. For any position $\phi$ on the spline, only $d{+}1$ basis functions are non–zero. For simplicity, we restrict $\phi$ to the range $[0,1]$ regardless of $K$ or $d$, and use cardinal B–splines to make knots equidistant in the

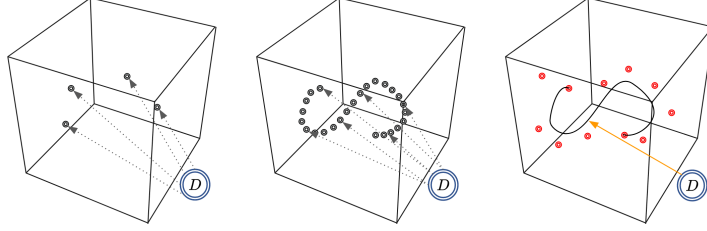

Figure 2: The Embedding Trick: (left) Discrete NDGs have a finite set of points to choose from. (middle) Increasing the number of nodes reveals that the weights lie on a lower dimensional manifold. (right) The embedding trick represents this manifold with a spline, and replaces the discrete decision making process with a continuous projection.

position space spanned by $\phi$. Note that they can still be anywhere in the actual space they live in. The degree of the splines also controls how far away the samples can see on the spline, helping with the exploration problem common with conditional architectures.

We apply the embedding trick to both transformation weights $\omega$ and decision parameters $\theta$ separately, so that a transformer spline $S_\omega^i$ generates the $\omega^i$ and a decision spline $S_\theta^i$ generates the $\theta^i$. The spline $S_\omega^i$ and its knots $C_{\omega,k}^i$ are in $\mathbb{R}^{g^i}$, and $S_\theta^i$ and its knots $C_{\theta,k}^i$ are in $\mathbb{R}^{h^i}$. Figure 3 shows the corresponding graphical models. The model on the left shows a dynamic, non-hierarchical SplineNet, and the model on the right is a hierarchical SplineNet. Here, the orange arrows correspond to the projections, blue arrows indicate transformations, red arrows are generation of transformation parameters and green arrows are generation of decision weights. The dashed orange arrow indicates a topological constraint, which is used to restrict access to sections of a spline depending on the previous position of the sample, for instance to simulate tree-like behaviour. This can enforce a semantic relationship between sections of consecutive splines, roughly corresponding to mutually exclusive subtrees for instance. Since $\phi^i$ is analogous to the node index in the discrete case, it can be used to determine the valid children nodes, which corresponds to a sub-range of the positions available on the spline of the next layer.

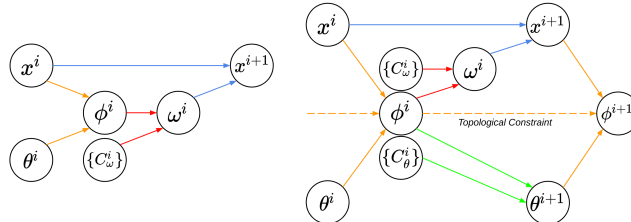

Figure 3: Graphical models with embedding trick: (left) Dynamic, non-hierarchical SplineNet (right) Hierarchical SplineNet. The additional dependency between positions $\phi^i$ and $\phi^{i+1}$ allows us to define topological constraints to simulate tree-like architectures.

Without any topological constraints, the projection becomes simply $\phi^i = D(x^i; \theta^i)$, where $D$ is a mapping $\mathbb{R}^{M^i} \to [0, 1]$, with $M^i = \dim(x^i)$ (orange arrows). In this work, we assume a linear projection (dense or convolutional) followed by a sigmoid function, but any differentiable mapping can be used. With topological constraints, we define the new position $\phi^{i+1}$ as a linear combination between the old position and the new projection:

$$\phi^{i+1} = (1 - \delta^i)\phi^i + \delta^i D(x^i; \theta^i). \tag{1}$$

Here, $\delta^i$ is a layer dependent *diffusion parameter* that controls the conditional range of projections. Setting $\delta^i = 1$ roughly simulates a decision jungle [18] with an infinite branching factor, which has no restrictions on how far samples can jump between layers. While regular decision jungles maintain only a small number of valid children for each node, SplineNet jungles allow transition to any one of the infinitely many nodes in the next layer. By setting $\delta^i = b^{1-i}$ we can also simulate a $b$-ary decision tree, but it has the issues with mini-batch size discussed earlier, and performs typically worse. A fixed $\delta^i = \alpha$ spans a novel continuous family of architectures for $0 \le \alpha \le 1$.

Note that while the topological constraint adds a flavor of hierarchical dependency to the network, a proper decision graph should pick different decision parameters for each node. This constraint determines the graph edges between consecutive layers, but it does not have an effect on the decision parameters.

## 2.2 SplineNet operators

**Projections** Projection $D$ is a mapping from feature map $x^i$ to spline position $\phi^i$. While any differentiable mapping is valid, we consider two simple cases: (i) a dot product with the flattened feature vector, and (ii) a $1 \times 1$ convolution followed by global averaging, which has fewer parameters to learn. In the first case, $\theta^i$ is a vector in $\mathbb{R}^{M^i}$, and the time and memory complexities are both $O(M^i)$. While this is negligible for networks with large dense layers, it becomes significant for all–convolutional networks. For the second case, $\theta^i$ is a filter in $\mathbb{R}^{1,1,c,1}$ where $c$ is the number of feature map channels. In both cases, a sigmoid with a tunable slope parameter is used to map the response to the right range.

**Decision parameter generation** The splines $S^i_\theta$ of the hierarchical SplineNets all have knots $C^i_{\theta,k}$ that live in the same space as $\theta^i$, which is $\mathbb{R}^{M^i}$. The cost of generating $\theta^i$ is then a weighted sum over $d+1$ such vectors, where $d$ is the degree of the spline. The time complexity is $O(dM^i)$, and the memory complexity is $O(K^i_\theta M^i)$, where $K^i_\theta$ is the number of knots.

**Fully connected layers** Fully connected layers have weights $\omega$ in the form of a matrix, such that $\omega^i \in \mathbb{R}^{M^i \times M^{i+1}}$. Thus, the corresponding spline must have knots that also lie in the same space. The additional costs compared to a regular fully connected layer besides the projection is the generation of $\omega$ via the spline function $S^i_\omega$, which has a time complexity $O(dM^iM^{i+1})$, and a memory complexity of $O(K^i_\omega M^iM^{i+1})$, where $K^i_\omega$ is the number of knots.

**Convolutional layers** Conventional 2D convolutional layers have rank-4 filter banks in $\mathbb{R}^{h,w,c,f}$ as weights $\omega$, where $h$ is the height and $w$ is the width of the kernels, $c$ is the number of input channels, and $f$ is the number of filters (we drop superscript $i$ for simplicity). We propose two different ways of representing filter banks with splines: (i) a single spline with rank-4 knots, or (ii) $f$ splines with rank-3 knots. While the first approach is a single curve between entire filter banks in $\mathbb{R}^{h,w,c,f}$, the second approach uses a spline per filter, each with knots living in $\mathbb{R}^{h,w,c}$. Rank-4 case is straightforward, with a single spline $S^i_\omega$ generating the filter bank as before. The additional time complexity is $O(dhwcf)$, and the memory complexity is $O(K^i_\omega hwcf)$. To handle multiple splines with rank-3 filters, we need to project to each spline separately. In the case of dot product decisions, this effectively turns $\theta^i$ to a matrix of form $\mathbb{R}^{f \times M^i}$, such that the projection $D(x^i; \theta^i) = \text{sigmoid}(\theta^i x^i)$ results in a $\phi^i \in \mathbb{R}^f$. Then, the final filter bank can be generated by:

$$\omega^i = \bigoplus_{j=1}^{f} S^i_{\omega,j}(\phi^i_j), \tag{2}$$

where $\phi^i_j$ is the $j$th element of $\phi^i$ that corresponds to the spline $S^i_{\omega,j}$, and $\bigoplus$ is the stacking operator that forms the filter bank. The additional time complexity is $O(M^i f + dhwcf)$ and the memory needed is $O(M^i f + K^i_\omega hwcf)$. Adapting the same approach to convolutional decisions is straightforward by changing the decision filter dimensionality to $\mathbb{R}^{1,1,c,f}$. When plugging many such rank-3 layers together, it may not be possible to diffuse positions coming from the previous layer with the new ones directly using Equation 1, since they may have different sizes. In such cases, we multiply the inherited positions with a learned matrix to match the shapes of position vectorsd.

## 2.3 Regularizing SplineNets

The continuous position distribution $P(\phi^i)$ plays an important role in utilizing SplineNets properly, *e.g.* if all $\phi^i$'s that are dynamically generated by samples are the same, the model reduces to a regular CNN. Figure 4 shows some real examples of such under-utilization. Meanwhile, we would also like to *specialize* splines to data, such that each control point handles a subset of classes or clusters more effectively. Both of these problems are common in decision trees, and the typical

solution is maximizing the mutual information (MI) $I(Y;\Lambda) = H(Y) - H(Y|\Lambda)$, where $Y$ and $\Lambda$ correspond to class labels and (discrete) nodes. In the absence of $Y$, $X$ can be used to specialize nodes to clusters.

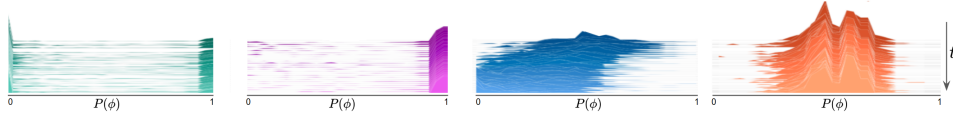

Figure 4: The distribution of $\phi^i$ can be suboptimal in various ways. Some examples directly taken from Tensorboard, where depth indicates training time: (1) binary positions, (2) constant positions, (3) positions slowly shifting, and (4) under-utilization. Ideally, the distribution $P(\phi^i)$ should be close to uniform.

In the case of SplineNets, the MI between the class labels and the continuous spline positions is:

$$I(\phi^i; Y) = H(\phi^i) - H(\phi^i|Y) = H(Y) - H(Y|\phi^i). \tag{3}$$

We choose to use the first form, which explicitly maximizes the position entropy $H(\phi)$, which we call the *utilization term*, and minimizes the conditional position entropy $H(\phi|Y)$, which is the *specialization term*. Then, our regularizer loss becomes:

$$\mathbb{L}^i_{reg} = -w_u H(\phi^i) + w_s H(\phi^i|Y), \tag{4}$$

which should be minimized. The $w_u$ and $w_s$ are utilization and specialization weights, respectively. To calculate these entropies, the underlying continuous distributions $P(\phi^i)$ and $P(\phi^i|Y)$ need to be estimated from the $N$ sampled position-label pairs $\{\phi^i_n, y_n\}^N_{n=1}$. The two most common techniques for this are Kernel Density Estimation (KDE) and quantization. Here, we describe a differentiable quantization method that can be used to estimate these entropies.

**Quantization method**   To approximate $P(\phi^i)$ and $P(\phi^i|Y)$, we can quantize the splines into $B$ bins and count samples that fall inside each bin. Normalizing the bin histograms can then give us probability estimates that can be used to calculate entropies. However, a loss based on entropy calculated with hard counts cannot be used to regularize the network, since the indicator quantization function is non-differentiable. To solve this problem, we construct a soft quantization function:

$$U(\Phi; c_b, w_b, \upsilon) = 1 - (1 + \upsilon^{(2(\Phi - c_b)/w_b)^2})^{-1}. \tag{5}$$

This function returns almost 1 when position is inside the bin described by the center $c_b$ and width $w_b$, and almost 0 otherwise. The parameter $\upsilon$ controls the slope. Figure 5 shows three different slopes used to construct bins.

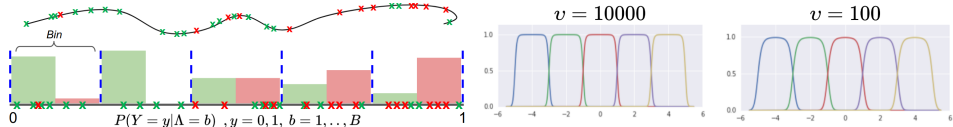

Figure 5: (left) Quantization method. (right) Five shifted copies of the soft quantization function $U$ with width=2 and slopes $\upsilon=\{10000, 100\}$.

We use this function to discretize the continuous variable $\phi^i$ with $B$ bins, which turns $\phi^i$ into the discrete variable $\Lambda^i$. Thus, the bin probabilities for $\Lambda^i = b$ and the entropy $H(\Lambda^i)$ become:

$$\Pr(\Lambda^i = b) \approx \frac{\sum_{n=1}^{N} U(\phi^i_n; c_b, w_b, \upsilon)}{\sum_{n=1}^{N} \sum_{b'=1}^{B} U(\phi^i_n; c_{b'}, w_{b'}, \upsilon)} \tag{6}$$

$$H(\Lambda^i) = -\sum_{b=1}^{B} \Pr(\Lambda^i = b) \log \Pr(\Lambda^i = b). \tag{7}$$

The specialization term is then:

$$\Pr(\Lambda^i{=}b|Y{=}c) \approx \frac{\sum_{n=1}^{N} U(\phi_n^i; c_b, w_b, v)\mathbb{1}(Y{=}c)}{\sum_{n=1}^{N}\sum_{b'=1}^{B} U(\phi_n^i; c_{b'}, w_{b'}, v)\mathbb{1}(Y{=}c)}, \tag{8}$$

$$H(\Lambda^i|Y) = -\sum_{c=1}^{C} \Pr(Y{=}c)\sum_{b=1}^{B} \Pr(\Lambda^i{=}b|Y{=}c)\log\Pr(\Lambda^i{=}b|Y{=}c). \tag{9}$$

The effect of using different values of $B$ on $P(\phi^i)$ can be seen in Figure 6. Using small $B$ creates artifacts in the distribution, which is why we use $B = 50$ in our experiments.

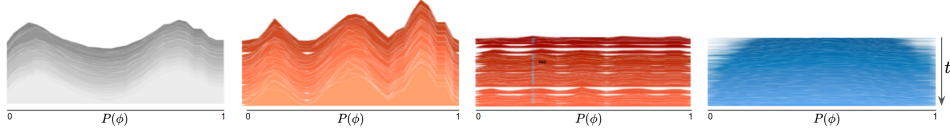

Figure 6: The effect of quantization based uniformization. From left to right: (i)$B = 2$, (ii)$B = 3$, (iii)$B = 5$, (iv)$B = 50$. We use $B = 50$ in this work.

The effect of specialization can be seen for a 10-class problem (Fashion MNIST) in Figure 7. Here, the top and bottom rows correspond to labeled sample distribution over splines for two consecutive layers, for four different values of $w_s$ (columns). Each image has 10 rows corresponding to classes, and 50 columns representing bins, where each bin's label distribution is normalized separately.



Figure 7: The effect of quantization based specialization. Increasing values of $w_s$ leads to more specialization.

## 3  Experiments

**Architecture**  SplineNets are generic and can replace any convolutional or dense layer in any CNN. To demonstrate this, we converted ResNets [19] and LeNets [20] into SplineNets. We converted all convolutional and dense layers into dynamic and hierarchical SplineNet layers, using different decision operators ('dot', 'conv') and knot types ('rank-3', 'rank-4'), and experimented with different number of knots. For ResNet, we relied on the public implementation inside the Tensorflow package with fine-tuned baseline parameters. We modernized LeNet with ReLu and dropout [21]. LeNet has two convolutional layers with $C_1$ and $C_2$ filters respectively, followed by two dense layers with $H$ and 10 hidden nodes. We experimented only with models where $C_1{=}s, C_2{=}2s, H{=}4s$ for various values of $s$ (depicted as LeNet-$s$). While the more compact mostly convolutional ResNet is better for demonstrating SplineNet's ability to increase speed or accuracy at the cost of model complexity, LeNet with large dense layers is better suited for showcasing the gains in model size and speed while maintaining (or even increasing) accuracy.

**Implementation details**  We implemented SplineNets in Tensorflow [22] and used stochastic gradient descent with constant momentum (0.9). We initialized all knot weights of a spline together with a normal distribution, with a variance of $c/n$, where $c$ is a small constant and $n$ is the fan in parameter that is calculated from the constructed operator weight shape. We found that training with conditional convolutions per sample using grouped or depthwise convolutions was prohibitively slow. Therefore we took the linear convolution operator inside the weighted sum over the knots, such that it applies to individual knots rather than their sum. This method effectively combines all

knots into a single filter bank and applies convolution in one step, and then takes the weighted sum over its partitions. Note that this is only needed when training with mini-batches; at test time a single sample fully benefits from the theoretical gains in speed.

**Hyperparameters**   A batch size of 250 and learning rate 0.3 were found to be optimal for CIFAR-10. Initializer variance constant $c$ was set to 0.05, number of bins $B$ to 50 and quantization slope $v$ to 100. We found it useful to add a multiplicative slope parameter inside the decision sigmoid functions, which was set to 0.4. The diffusion parameter had almost no effect on the shallow LeNet, so $\alpha$ was set to 1. With ResNet, optimal diffusion rate was found to be 0.875, but the increase in accuracy was not significant. Regularizer had a more significant effect on the shallow LeNet, with $w_s$ and $w_u$ set to 0.2 giving the best results. On ResNet, the effect was less significant after fine tuning the initialization and sigmoid slope parameters. Setting $w_s$ and $w_u$ to 0.05 gave slightly better results.

**Spline-ResNet**   We compared against 32 and 110 level ResNets on CIFAR-10, which was augmented with random shifts, crops and horizontal flips, followed by per-image whitening. We experimented with model type $M \in \{D, H\}$ (D is dynamic-only, H is hierarchical), decision type $T \in \{C, D\}$ (C is 1×1 convolution, D is dense), and the knot rank $R \in \{3, 4\}$. The model naming uses the pattern $M(K)$-$T$-$R$; *e.g.* D(3)-C-R4 is a dynamic model with three rank-4 knots and convolutional decisions. All SplineNets have depth 32. Baseline ResNet-32 and ResNet-110 reach 92.5% and 93.6% on CIFAR-10, respectively.

First experiment shows the effect of increasing the number of knots. For these tests we opted for the more compact convolutional decisions and the more powerful rank-3 knots. Figure 8 shows how the accuracy, model size and runtime complexity for a single sample (measured in FLOPS) are affected when going from two to five knots. Evidently, SplineNets deliver on the promise that they can increase model complexity without affecting speed, resulting in higher accuracy. For this setting, both the dynamic and hierarchical models reach nearly 93.4%, while being three times faster than ResNet-110.

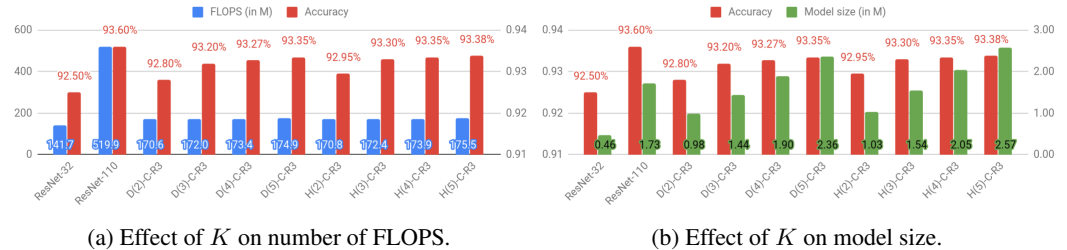

(a) Effect of $K$ on number of FLOPS.     (b) Effect of $K$ on model size.

Figure 8: Spline-Resnet-32 using convolutional decisions and rank-3 knots, with $K=2-5$. (Left) Increasing the number of knots increases the accuracy significantly, and has a negligible effect on the number of FLOPS for single sample inference. (Right) Model size grows linearly with number of knots.

Next, we compare all decision and convolution mechanisms for both architectures by fixing $K=5$. The results are given in Figure 9. Here, the most powerful SplineNet tested is H(5)-D-R3, with hierarchical, dense projections for each filter, and it matches the accuracy of ResNet-110 with almost half the number of FLOPS. However, a dot product for each filter in every layer creates a quite large model with 42M parameters. Its dynamic-only counterpart has 10M parameters and is three times faster than ResNet-110. With close to 93.5% accuracy, H(5)-D-R4 provides a better trade off between model size and speed with 3.67M parameters. Note that we have not tested more than five knots, and all models should further benefit from increased number of knots.

**Spline-LeNet**   We trained LeNet-32, LeNet-64 and LeNet-128 as baseline models on CIFAR-10. We used SplineNets with the same parameters of LeNet-32, combined with the more powerful rank-3 knots and dot product decisions. The comparisons in accuracy, runtime complexity and model size are given in Figure 10. Notably, LeNet model size increases rapidly with number of filters, with a large impact on speed as well. Increasing the number of knots in SplineNets is much more efficient in terms of both speed and model size, leading to models that are as accurate as the larger

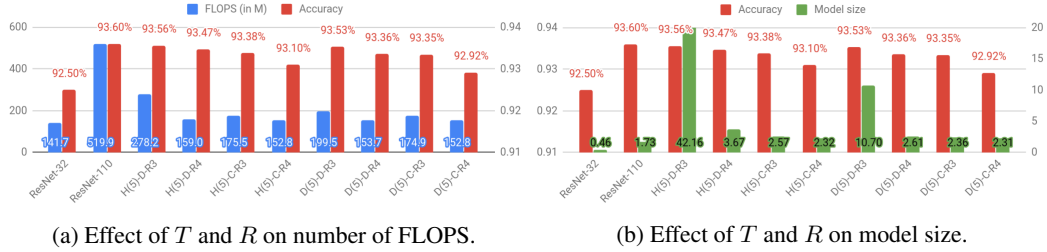

(a) Effect of $T$ and $R$ on number of FLOPS.

(b) Effect of $T$ and $R$ on model size.

Figure 9: Dynamic and Hierarchical Spline-Resnets with 32 levels and 5 knots, with different options for decision types and knot ranks.

baseline, while also being 15 times faster and almost five times smaller. These results show that SplineNets can indeed reduce model complexity while maintaining the accuracy of the baseline model, especially in the presence of large dense layers.

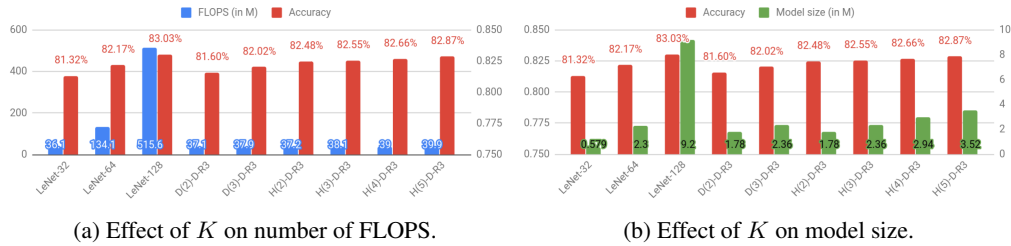

(a) Effect of $K$ on number of FLOPS.

(b) Effect of $K$ on model size.

Figure 10: Dynamic and Hierarchical Spline-LeNets with $K=2-5$, using rank-3 knots and dot product decisions.

Finally, we experimented with Spline-LeNets on MNIST. We augmented MNIST with affine and elastic deformations and trained LeNet-32 as a baseline, which achieved 99.52% on the original, and 99.60% on the augmented dataset. In comparison, H(2)-D-R3 reached 99.61% and 99.65% on the respective datasets. The best score of 99.71% was achieved by H-SN(4) on the augmented dataset with 2.25M parameters. In comparison, CapsuleNets [23] report a score of 99.75% with 8.5M parameters. Higher scores can typically only be reached with ensemble models.

## 4    Conclusions and Discussions

In this work, we presented the concept of SplineNets, a novel and practical method for realizing conditional neural networks using embedded continuous manifolds. Our results dramatically reduce runtime complexity and computation costs of CNNs while maintaining or even increasing accuracy.

Like other conditional computation techniques, SplineNets have the added benefit of allowing further efficiency gains through *compression*, for instance by pruning layers [24, 25] in an energy-aware manner [26], by substitution of filters with smaller ones [27], or by quantizing [28, 29, 30] or binarizing [31] weights. While these methods are mostly orthogonal to our work, one should take care when sparsifying knots independently.

We consider several avenues for future work. The theoretical limit of gains in accuracy from increasing knots is not clear from the results so far, for which we will conduct more experiments. Also there is a large jump in accuracy and model size when switching from rank-4 to rank-3 knots. To investigate the cases between the two, we will introduce the concept of spline groups, where groups contain rank-3 knots and projections need to be per group rather than per filter. Another interesting direction is using SplineNets to form novel hierarchical generative models. Finally, we will explore methods to jointly train SplineNet ensembles by borrowing from the decision tree literature.

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
