[Reviews · NeurIPS 2018]

Reviewer 1



This paper presents a new type of neural network where the transformation of each layer is conditioned on its input and its “depth” (computation path), generated from a very compact set of parameters. The network also can be viewed as a continuous generalization of neural decision graphs. In my opinion, this is a novel and creative work. The B-spline embedding trick is clever and efficient. The model sizes and runtime operations compared to those of CNNs of similar accuracies are truly impressive. Overall, this is an accept. My suggestions for the authors to improve their (already good) paper are as follows. 1) Test on more recent CNN architectures (e.g. VGG, if ResNet is not feasible). 2) Include SOTA results (accuracies and model sizes) under all datasets (it’s okay for a novel work to be not always better). 3) Consider using more datasets (e.g. CIFAR100 & SVHN, if ImageNet is not feasible). 4) Consider using existing methods to regularize the utilization of the splines (e.g. normalization), instead of adding an extra technique (differentiable quantization) to the already dense manuscript. (After rebuttal) I appreciate the authors’ feedback and have updated my score accordingly.

Reviewer 2



The paper presents SplineNets that reformulate CNNs as neural decision graph using B-splines. It's comprised of four technical contributions, i.e., embedding trick, general formulation for a neural decision graph, a loss function of utilizing and specializing splines and a differentiable quantization method. The idea is novel and sensible. It integrates the classic splines into convolutional neural networks which might be valuable for both of theoretical and practical aspects. However, the paper is difficult to read and follow. It is unclear how they implement in practice. Some equations and symbols need to be clarified, e.g. the symbols in equation 7-10, and d indicates degree of the polynomials in line 115 but hyperparameter delta in line 235. I'm confused with several points. a) what's the role of theta? Is the phi (i.e. D(x; theta) in line 137) analogous to attention (or gating) mechanism? b) Does the k/K in line 233 denote the knot number? Does the degree of the polynomial default to be k-1? Or is a fixed degree used? If larger K indicates higher-order smoothness in F-SN, does it mean linear spline works best as J-SN always works better than F-SN. b) how to demonstrate the dynamic (conditioned on the input) introduced by the method? d) More experimental details need to be provided and clarified, e.g. optimization method, important hyperparameters for training, filter sizes. I like the idea, but the current draft need to be improved. After reading the comments of other reviewers and rebuttal, I think it's a good work. I hope the final manuscript could include the updates present in the rebuttal.

Reviewer 3



In this paper, the authors introduce SplineNet to learn the conditional NNs. The authors propose a embedding trick to learn the embedded mainfolds and a regularized loss to encourage the maximum information gain. Through experiments on MNIST and CIFAR10, the authors demonstrate SplineNet achieve comparable performance with baselines with much less computation and parameters. The idea is novel and the proposed method is sound. The ablative study and experimental analysis is helpful to better understand the working principles of the method. My questions on the experiments: --The authors only conduct experiments on small scale datasets (MNIST/CIFAR) using shallow networks (LeNet). How does it perform on larger datasets such as ImageNet using more complex networks such as DenseNet/ResNet. --Are the speed-up ratios presented in L249 theoretical or practical? ============================ Thanks for the feedback. I suggest the authors add the new experimental results in the final version.